# Can Neutrophils Prevent Nosocomial Pneumonia after Serious Injury?

**DOI:** 10.3390/ijms24087627

**Published:** 2023-04-21

**Authors:** Kristína Macáková, Elzbieta Kaczmarek, Kiyoshi Itagaki

**Affiliations:** 1Department of Surgery, Beth Israel Deaconess Medical Center/Harvard Medical School, Boston, MA 02215, USA; kristinmacakova@gmail.com (K.M.); kitakaczmarek@gmail.com (E.K.); 2Institute of Molecular Biomedicine, Faculty of Medicine, Comenius University, Sasinkova 4, 811 08 Bratislava, Slovakia

**Keywords:** neutrophils, injury, trauma, innate immunity, infection, nosocomial pneumonia, FPR1

## Abstract

Nosocomial pneumonia is a leading cause of critical illness and mortality among seriously injured trauma patients. However, the link between injury and the development of nosocomial pneumonia is still not well recognized. Our work strongly suggests that mitochondrial damage-associated molecular patterns (mtDAMPs), especially mitochondrial formyl peptides (mtFPs) released by tissue injury, play a significant role in developing nosocomial pneumonia after a serious injury. Polymorphonuclear leukocytes (neutrophils, PMN) migrate toward the injury site by detecting mtFPs through formyl peptide receptor 1 (FPR1) to fight/contain bacterial infection and clean up debris. Activation of FPR1 by mtFPs enables PMN to reach the injury site; however, at the same time it leads to homo- and heterologous desensitization/internalization of chemokine receptors. Thus, PMN are not responsive to secondary infections, including those from bacteria-infected lungs. This may enable a progression of bacterial growth in the lungs and nosocomial pneumonia. We propose that the intratracheal application of exogenously isolated PMN may prevent pneumonia coupled with a serious injury.

## 1. Introduction

### 1.1. A Serious Risk of the Development of Pneumonia after Trauma

Trauma is the leading cause of death in people under age 45 and the third death cause overall, with nosocomial pneumonia being accountable for morbidity in trauma patients [1,2,3,4]. Thus, preventing nosocomial pneumonia would significantly reduce morbidity, mortality, and hospital costs. Additionally, intubated injured patients suffer pneumonia far more often than uninjured intubated patients do [5]. However, the mechanistic links between injury and nosocomial pneumonia are mostly unknown. Our recent data confirm a strong association between tissue injury and nosocomial pneumonia and suggest a causal relationship with limited PMN migration to the lung under extrapulmonary injury conditions.

### 1.2. Mitochondrial DAMPs Released from Injured Tissues Predispose to Pneumonia Due to Reduced PMN Recruitment to the Lungs

Mitochondria are widely recognized to originate as bacterial endosymbionts [6]. They are hidden in the cells from immune recognition and produce energy as adenosine triphosphate (ATP). Still, when released from injured tissues, they present bacteria-specific molecular motifs, including N-formylated peptides (mtFPs) and mitochondrial DNA (mtDNA) that can act as DAMPs capable of modulating systemic immune responses as if bacterial infection happened in the body [7,8,9,10,11,12]. Mitochondria contain 13 mtFPs [13], which are similar to N-formyl peptides derived from bacterial proteins and hence may act as powerful PMN chemoattractants via FPR1 [13]. We have shown that exposure to mtFPs originating from clinical injuries renders PMN less sensitive to lung-derived chemokines due to chemokine receptor desensitization/internalization, thus decreasing PMN migration and PMN recruitment to the lungs after bacterial inoculation [7,14,15,16]. These events correlate with attenuated bacterial clearance as shown in the lungs of injured mice [12]. Therefore, activation of PMN FPR1 by mtFPs released from injury sites is a critical factor in the potentiation of post-traumatic pneumonia.

### 1.3. Novel, Non-Antibiotic, Cell-Based Therapies May Prevent Pneumonia after Trauma

Bacteria are increasingly resistant to available antibiotics, with some strains resistant to most or all available treatments. This includes *S. aureus*, a common pathogen responsible for pulmonary infections after trauma [17,18,19,20]. Antibiotic-resistant infections may cause severe morbidity and mortality and higher healthcare costs [21]. Moreover, new antibiotics are not expected to be readily available in the near future [22,23]. If one of the defense mechanisms against pneumonia after an injury is missing, such as exhaustion of PMN function due to their partial activation, direct application of functionally active PMN to the airway may have a therapeutic value. Therefore, we propose studying such a novel, cell-based therapy and believe it could become a new form of prevention of nosocomial pneumonia in trauma patients.

### 1.4. Granulocyte Transfusion

We expect that this is feasible as granulocyte transfusion (iv) has been performed at major hospitals worldwide for neutropenic patients, including cancer patients after chemotherapy [24]. Granulocytes must be used within 24 h after isolation while still viable. They are frequently contaminated with red blood cells (RBS), T-cells, and other cells. Thus, ABO-matched granulocytes are irradiated before transfusion to reduce T-cell content, which could lead to transfusion-associated graft-versus-host disease (TA-GVHD). Irradiation prevents T-cells proliferation after transfusion but does not affect PMN that do not proliferate and die within ~24 h [25].

No human leukocyte antigen (HLA) is examined before transfusion. Thus, repeated PMN transfusions from various donors will decrease the number of PMN in circulation after transfusion due to anti-HLA antibody production from previous donors’ PMN. It also strictly suggests that clinicians weigh the danger of transfusion vs. its benefits to the patients [24,26] since it is not a well-established procedure and can be fatal to the recipients. Thus, if the pure PMN can be isolated and stored, these could be used for granulocyte transfusion that requires more frequent and increased PMN.

## 2. Our Vision: Application of PMN to Human Lungs Will Prevent/Cure Nosocomial Pneumonia after Injury

### 2.1. What Kind of PMN Can Be Applied Safely to the Lungs of Seriously Injured Trauma Patients to Prevent Nosocomial Pneumonia?

There are no reports on PMN airway instillation as a clinical treatment. Therefore, the safety of this procedure has to be investigated. We plan to examine what kind of PMN can be used safely. Moreover, we will establish the best methods of PMN isolation and storage before application. If the results are promising, we propose to use PMN as a cell therapy for trauma patients.

#### 2.1.1. Any PMN Isolated from Human Donors Will Work?

Many reports suggest that whole blood transfusion (A, B, O-matched) does not cause any issues for recipients [27,28]. It means that blood cells, including PMN, may cause no negative reaction and that HLA match may not be a hampering factor. Unlike other leukocytes, PMN are short-lived cells [29]. Therefore, after instillation they can kill bacteria in the lungs via phagocytosis and neutrophil extracellular traps (NETs) and die shortly after and phagocytosed by the recipients’ macrophages [30]. Again, high purity is the key to avoiding above-mentioned TA-GVHD. How can we prove that any PMN should work without causing any adverse effects on the recipients? Can PMN be purified in a hospital setting at high purity? Maybe not. If there is a simple system to isolate human PMN from peripheral blood at high purity, these PMN can be applied to patients.

#### 2.1.2. Autologous PMN

The trauma patient’s own PMN could be isolated and re-applied. It is well documented that the number of PMN in trauma patients is higher than in healthy people, depending on injury severity [31]. However, it is also well known that the functions of trauma patients’ PMN are reduced compared to those of healthy volunteers [25]. PMN’s migration, phagocytosis, reactive oxygen generation, and NETs formation are significantly decreased by trauma [32,33,34,35,36,37]. Reversing the trauma patients’ PMN to the functional cells quickly would make them safe to apply to the patients even with limited purity. Again, at this time, there is no system to isolate a high purity of PMN at hospitals.

#### 2.1.3. PMN Modified to Be Negative for HLA

As HLAs are the significant components recognized by recipients’ immune cells, PMN from HLA-disrupted iPS cells could be used for instillation. Our collaborator has established the methods of preparation of such cells [38], and these cells are investigated by our group as potential donor cells. We are evaluating ex vivo functions of PMN developed from HLA(-)iPS cells. We will then examine these PMN in our mouse model of injury and lung infection model to compare to freshly isolated human PMN for the ability to kill bacteria in the lungs after injury.

### 2.2. How to Store PMN before Applying Them to Trauma Patients

#### Cryopreservation of Human PMN

PMN are well known to be fragile after freeze–thaw in osmotically stressed hypertonic media [39]. It would be best if we could keep PMN frozen and thaw them when needed. No established method has been reported that shows high viability and normal functions of PMN after freeze–thaw. Recently, the freezing media CryoStor CS10 from Stem Cell Technologies showed very promising results. Frozen human PMN are available in CS10 media from Stem Cell Technologies. This is very promising. We are evaluating the viability and functions of PMN after freeze–thawing in CS10 using our volunteers’ PMN, which are essential for successful PMN storage. Once established, the successful freeze–thaw of a large number of pure HLA(-)PMN will benefit patients who might develop nosocomial pneumonia after injury but also who are neutropenia after chemotherapy [40,41], who are severe congenital neutropenia [42], and who do not have less functional PMN such as elderly people [43]. Thus, the application of PMN can be useful for wide variety of people.

## 3. Supporting Data

### 3.1. Mouse Injury and Lung Infection Model 

To mimic sterile, typical abdominal extrapulmonary injury in humans, we instilled mtDAMPs intraperitoneally (i.p.). This injury decreased the number of PMN migrating toward secondary infection in the lungs, thus bacterial clearance was reduced compared to control mice [7,14,44].

### 3.2. Prevention of Early Onset of Pneumonia

First, we mimicked the early onset of pneumonia with *Staphylococcus aureus* (SA). CON (control, non-injury, infected) and two injury groups were injected with mtDAMPs. Then, all mice were infected with SA in the lungs. One injury group received PMN from the same strain of mice to the lungs as a treatment (mtDAMPs + PMN). A day later, lung bacteria were examined compared to CON mice. Injury without treatment mice (mtDAMPs) showed the greatest number of SA in the lungs. CON and injury and PMN-treated mice showed better SA clearance in the lungs. Data suggest that injury reduced SA clearance in the lungs likely due to limited PMN migration toward the lungs. However, exogenous PMN application to the lungs increased bacterial clearance in the lungs (Figure 1) [14].

### 3.3. Prevention of Late-Onset Pneumonia

Next, we mimicked the late onset of pneumonia with *Pseudomonas aeruginosa* (*P. aeruginosa,* PA). As shown in Figure 1, animals were divided into three groups, CON, injury, and injury followed by PMN treatment. As a late onset of pneumonia, we infected animals later than above and applied PMN simultaneously after infection. Similar to the early onset of pneumonia, the late onset of pneumonia was significantly reduced by PMN application (Figure 2) [14].

### 3.4. Prevention of Late Onset of Pneumonia with an Increased Number of P. aeruginosa

We applied an increased number of PA. The rest are the same as in Figure 2. Even the CON group could not clear the bacteria well, so there is no significant difference between CON and injury alone groups. However, the application of PMN (mtDAMPs + PMN) significantly reduced lung bacteria (Figure 3) [14].

### 3.5. Evaluation of Potential Lung Injury Due to PMN Instillation

Although PMN application to the lungs effectively cleared secondary bacterial infection to the lungs after injury, we have to confirm the application will not damage the recipients’ lungs. We collected bronchoalveolar lavage fluid (BALF) after various treatments. For example, “CD-1 to BL6” means PMN from CD-1 were applied to the lungs of BL6 mice. Cross-strain application of PMN did not cause lung damage compared to CON mice, and mtDAMPs application did not cause any. mtDAMPs followed by bacteria and PMN or HBSS buffer increased lung damage significantly compared to other groups. However, PMN did not have any effect compared to HBSS. Thus, bacterial infection caused the lung damage detected by protein concentrations in BALF (Figure 4) not PMN application [14].

### 3.6. Human PMN Clears Bacteria in the Lungs without Causing Adverse Effects on the Recipients

Finally, we applied human PMN to our mouse model described in Figure 1 to determine whether human PMN can clear bacteria in the lungs without damaging lungs. CON- and PMN-treated group showed significantly increased bacterial clearance in the lungs (Figure 5) [44]. We also observed mice for up to 28 days after the application of human PMN. We examined the long-term effect of lung damage after applying human PMN via histology with H&E staining, and Trichome and Sirius Red staining to evaluate lung fibrosis. We found no lung damage or lung fibrosis. Additionally, human PMN was detected a day after application but not detected in BALF after 48 h by flow cytometry (CD16(+)/CD49d(-)).

### 3.7. Functions of Freeze–Thaw PMN

Recently, we found freeze–thaw PMN exhibited reactive oxygen species (ROS) production by stimulation of 100 nM fMLF and 100 nM PMA. Figure 6A shows real-time relative ROS production. Figure 6B shows ROS production by area under curve (AUC) [45]. PMA stimulation induced a lot of ROS production compared to fMLF. Compared to medium, fMLF and PMA induced significantly larger ROS production. ROS production by freeze–thaw PMN was significantly reduced compared to that of freshly isolated PMN. Then, we applied freshly isolated PMN and freeze–thaw PMN to our mouse injury and bacterial infection model described above. Preliminary data suggested that freeze–thaw PMN application enhanced the bacterial clearance in the lung as much as freshly isolated PMN even though there was no significant difference compared to mtDAMPs/bacterial infection group (Figure 7). These are encouraging data that need further examination. So far, the application of human PMN did not cause any damage to the recipient mice. This is not surprising since human PMN are known to have a very short life [29]. They kill bacteria after application and die and then are phagocytosed by residential macrophages. Advancing this method toward possible human use requires study in a large-animal model. Pigs are recognized as an appropriate animal model for studying infectious diseases since their immune system shares many structural and functional similarities with humans [46]. Moreover, pig lungs and airways are anatomically similar to humans [47].

## 4. Summary

So far, we have shown that the application of mouse PMN of different strains or even human PMN can help clear lung bacteria after injury using our established mouse models without causing any adverse effects on recipients. Up to 28 days after human PMN application, we could not detect any lung injury, including fibrosis, mainly due to the short life of PMN. Though the instillation of exogenous PMN into the lung seems contradictory since the patients with pneumonia already have many PMN in the lungs, these instilled PMN will be activated by bacteria and will damage various tissues, including the lungs. This may not be the case. We believe “the timing is everything”. The lungs of established pneumonia patients have many PMN. The important point is when PMN migrated to the lungs. PMN could not migrate to the lungs when the bacterial infection started due to receptor desensitization as a result of the increased amount of circulating mtDAMPs. PMN could migrate to the lungs only after overwhelming bacterial growth, pneumonia was established, and mtDAMPs in circulation finally decreased. Applying PMN at the early stage of bacterial infection is the key to killing them. No one has ever examined the changes in the number of PMN in the lungs of trauma patients who gradually developed nosocomial pneumonia. We think PMN will not migrate to the lungs after serious injury compared to the uninjured patients. Later, we find many PMN even from trauma patients who developed pneumonia. Recently, Paul Kubes and others suggested that PMN will repair many organs even though they may damage at the beginning when they need to phagocytose cellular debris and bacteria after injury and infections [48]. PMN even contribute to angiogenesis and dying/being phagocytosed by macrophages. Thus, an increasing number of new findings support that additional PMN will do good.

## 5. More toward Human

It seems that intratracheal application of various types of PMN including human origin did not cause any adverse effects to the recipient mice and effectively cleared bacteria in the lungs. However, the question is, “Is it really so in the case of humans?” PMN live only for a short time, thus the applied PMN will be cleared from the recipients without/before causing any issues. This could be true. Our method is being applied to the pig models under the same concept for the next step. This is an important step toward human use as described before. On the other hand, “Is HLA match an important factor?” PMN without HLA will be the perfect candidates for safe PMN application when they are proven to function similarly to freshly isolated PMN and can maintain functions after freeze–thaw. Once established, these PMN can be used even for neutropenic patients that require many PMN to be infused to prevent infection.

## 6. Detailed Methods

For readability and nature of this article, we list detailed methods here with references. Here, we list selected papers that will help understand our vision.

Human PMN preparation [49], mouse BM-PMN preparation [50], detailed mouse injury/lung infection models [14,44], mitochondrial DAMPs [7,11,51].

## Figures and Tables

**Figure 1 ijms-24-07627-f001:**
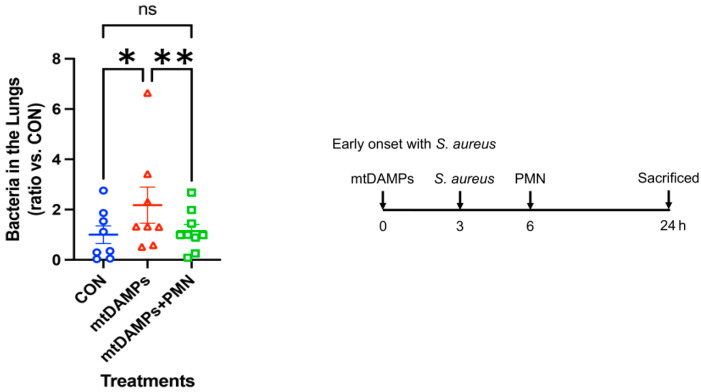
Effects of mtDAMPs and external mouse PMN on *S. aureus* (SA) clearance in lung. **Left:** CD-1 mice were separated into three groups. 1. **CON:** Saline i.p. injection (time 0) followed by *S. aureus* (3 h) i.t. 2. **mtDAMPs:** mtDAMPs i.p. followed by *S. aureus*, 3. **mtDAMPs + PMN:** mtDAMPs i.p. followed by *S. aureus* and BM-PMN i.t. (6 h). Animals were sacrificed t = 24 h. Lung homogenates were prepared to determine bacterial presence. mtDAMPs prepared from 10% of total liver in saline. 50 µL of OD600 = 0.1 *S. aureus* was injected intratracheally. BM-PMN were freshly prepared from donor CD-1 mice and ~2 × 10^6^ cells were injected intratracheally. The numbers of animals used for CON, mtDAMPs, and mtDAMPs + PMN were N = 8, N = 8, and N = 9, respectively. Mean and SE values are shown. * denotes a significant difference by one-way ANOVA, Tukey [14]. *: *p* = 0.027, **: *p* = 0.015. ns: not significant. **Right**: The animal protocol as early onset with *S. aureus*.

**Figure 2 ijms-24-07627-f002:**
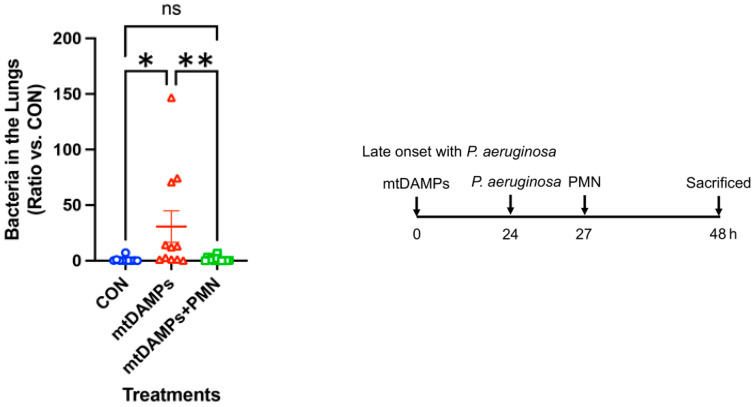
Effects of mtDAMPs and external mouse PMN on *P. aeruginosa* clearance (PA) in lung. **Left:** CD-1 mice were separated into three groups. 1. **CON:** Saline i.p. injection (time 0) followed by *P. aeruginosa* (24 h) i.t. 2. **mtDAMPs:** mtDAMPs i.p. (time 0) followed by *P. aeruginosa* i.t. (24 h), 3. **mtDAMPs + PMN:** mtDAMPs i.p. (time 0) followed by *P. aeruginosa* i.t. (24 h) and BM-PMN i.t. (27 h). Animals were sacrificed t = 48 h. Lung homogenates were prepared to determine bacterial presence. mtDAMPs were prepared from 10% of total liver in saline. A total of 50 μL of OD600 = 0.1, *P. aeruginosa* was injected intratracheally. BM-PMN were freshly prepared from donor CD-1 mice and ~2 × 10^6^ cells were injected intratracheally. The numbers of animals used for CON, mtDAMPs, and mtDAMPs + PMN were N = 10, N = 11, and N = 12, respectively. Mean and SE values are shown. * denotes a significant difference by one-way ANOVA, Tukey [14]. *: *p* = 0.047, **: *p* = 0.038. ns: not significant. **Right**: The animal protocol as late onset with *P. aeruginosa*.

**Figure 3 ijms-24-07627-f003:**
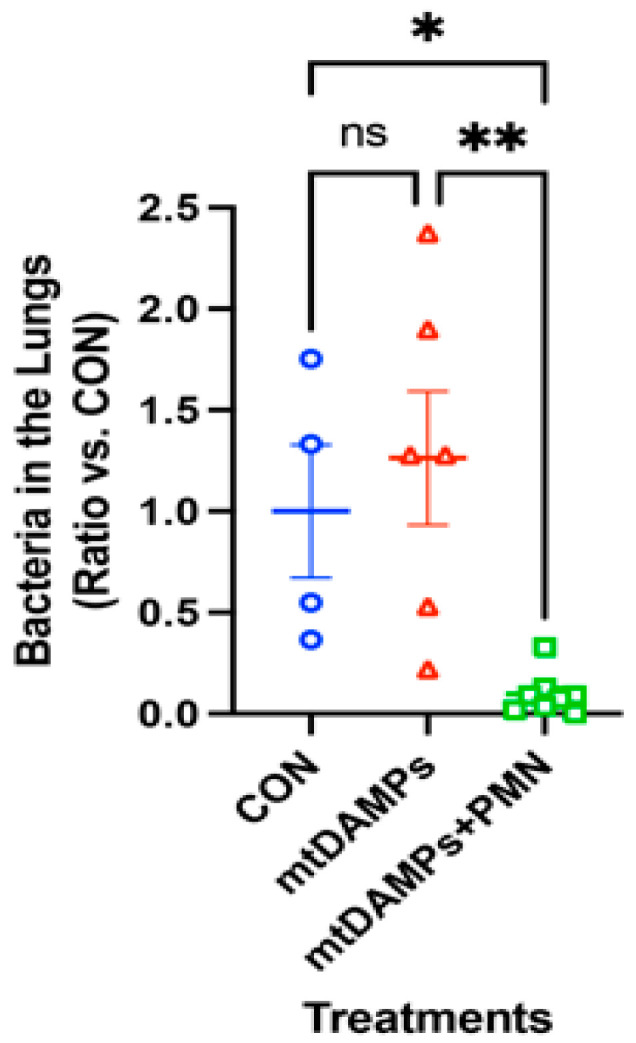
Effects of exogenous PMN on clearance of *Pseudomonas* pneumonia. Protocols are similar to Figure 2; however, an increased number of *P. aeruginosa* (OD = 0.111) was injected to the lungs. The numbers of animals used for CON, mtDAMPs, and mtDAMPs + PMN were N = 4, N = 6, and N = 8, respectively. Mean and SE values are shown. *: denotes a significant difference by one-way ANOVA, Tukey [14]. *: *p* = 0.036, **: *p* = 0.004. ns: not significant.

**Figure 4 ijms-24-07627-f004:**
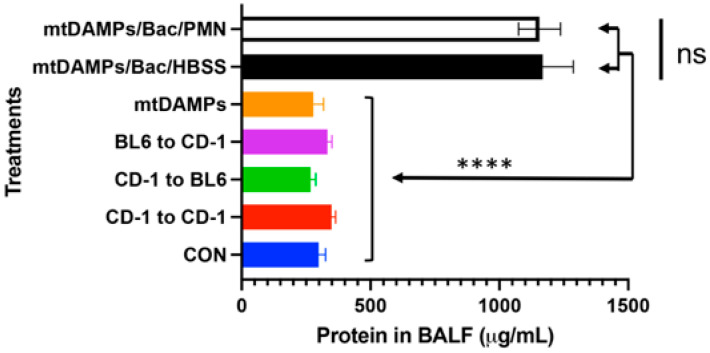
Exogenous PMN do not injure the lung. Lung injury was evaluated by assaying protein leak into BAL fluids. **CON:** saline i.t. (n = 5), **CD-1 to CD-1:** BM-PMN from CD-1 mice were instilled i.t. into CD-1 mice, 2 × 10^6^ i.t. (n = 5), **CD-1 to BL6:** BM-PMN from CD-1 mice were instilled i.t. into BL6 mice (n = 2). **BL6 to CD-1:** BM-PMN from BL6 mice (n = 2) were instilled i.t. into CD-1 mice. **mtDAMPs/Bac/HBSS**: mtDAMPs from 10% liver is given i.p. at t = 0. *S. aureus* is given i.t. (8.6 × 10^6^ CFU) at T = 3 h and followed by HBSS (vehicle for PMN) (n = 3). **mtDAMPs/Bac/PMN**: As in mtDAMPs/Bac but followed by PMN i.t. (1 × 10^6^) at T = 6 h (n = 3). *: denotes a significant difference by one-way ANOVA with Tukey’s test. ns denotes *p* = 0.090. ****: *p* < 0.0001. ns: not significant. Sample collection: Control (saline) and CD-1 to CD-1 PMN i.t. 24 h, BL6 to CD-1 or CD-1 to BL6: 72 h, mtDAMPs/Bac/HBSS, and mtDAMPs/Bac/PMN: 23 h. The number in bracket represents the number of animals used [14].

**Figure 5 ijms-24-07627-f005:**
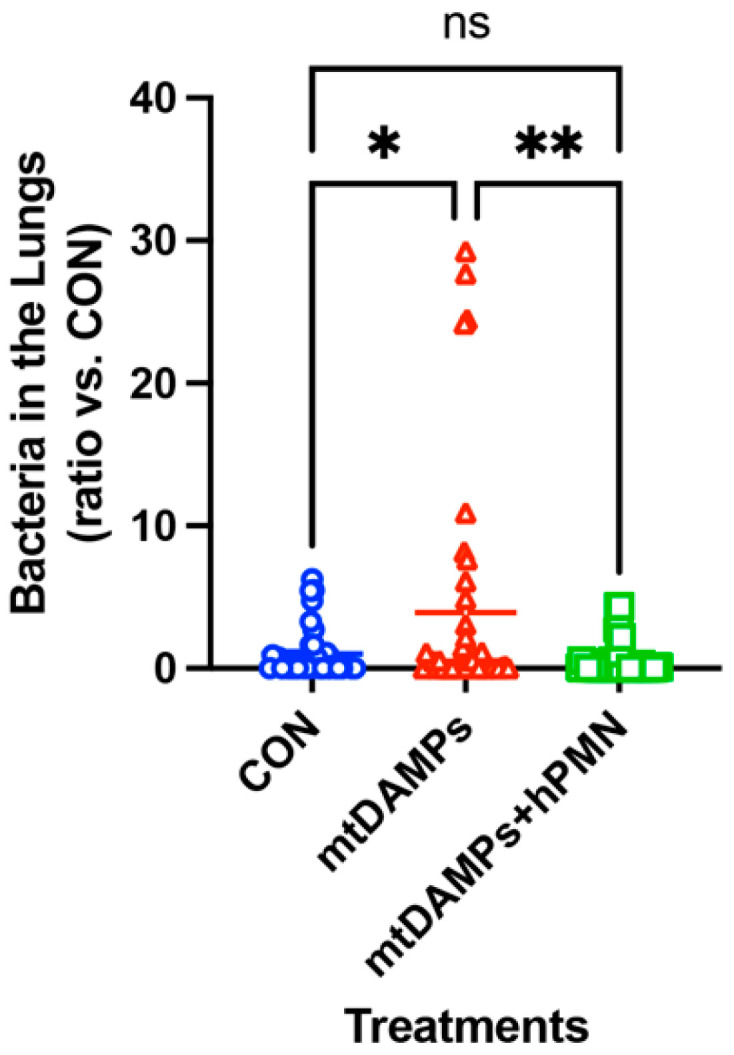
Effects of mtDAMPs and external human PMN on *S. aureus* (SA) clearance in lung. C57BL6 mice were separated into three groups. Protocols are similar to Figure 1. **1**. **CON:** Saline i.p. injection (time 0) followed by *S. aureus* (3 h) i.t. **2**. **mtDAMPs:** mtDAMPs i.p. (time 0) followed by *S. aureus* i.t. (3 h) and saline i.t. (6 h). **3. mtDAMPs + hPMN:** mtDAMPs i.p. (time 0) followed by *S. aureus* i.t. (3 h) and human PMN i.t. (6 h). Animals were sacrificed t = 20 h. Lung homogenates were prepared to determine bacterial presence. MTD were prepared from 10% of total liver in saline. A total of 50 μL of OD600 = 0.1, *S. aureus* was applied intratracheally. Human PMN were freshly prepared from healthy donor and ~2 × 10^6^ cells were injected intratracheally. The numbers of animals used for CON, mtDAMPs, and mtDAMPs + PMN were n = 18, n = 21, and n = 21, respectively. Mean and SE values are shown. *: denotes a significant difference by one-way ANOVA, Tukey [44]. *: *p* = 0.0129, **: *p* = 0.0020. ns: not significant.

**Figure 6 ijms-24-07627-f006:**
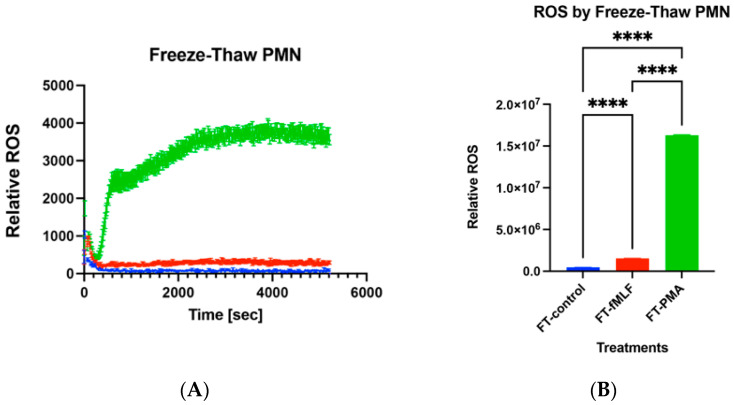
Reactive oxygen species production by freeze–thaw PMN. Freeze–thaw human PMN were loaded with Luminor and 100 nM fMLF (red), PMA (green), or buffer (blue) were applied to stimulate PMN to produce reactive oxygen species (ROS). Real time ROS production is shown in (**A**). Area under curve (AUC) for 5200 s was calculated to compare the ROS production (**B**). Experiments were done in quadruplicates. Mean and SE values are shown. *: denotes a significant difference by one-way ANOVA, Tukey. ****: *p* < 0.0001.

**Figure 7 ijms-24-07627-f007:**
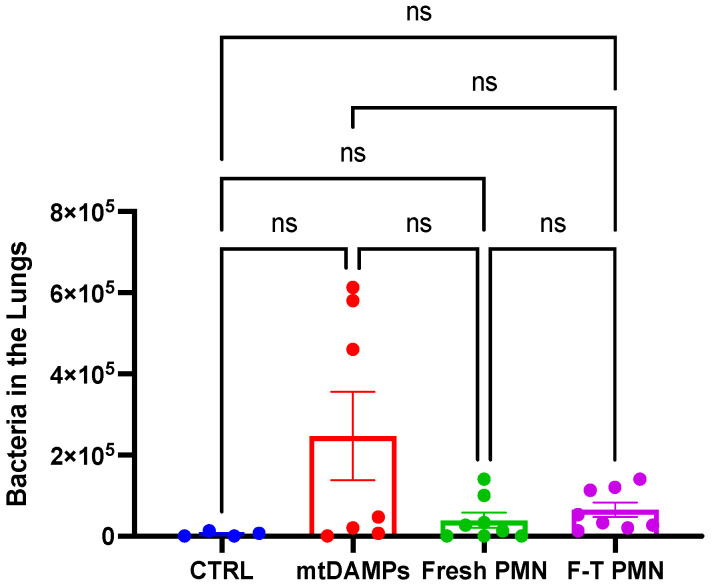
Freeze–thaw PMN induce bacterial killing in vivo. Similar to Figure 5, we applied freshly isolated and freeze–thaw human PMN to our mouse injury/lung bacterial infection model. Although there was no significant difference (ns), we could see a tendency that freeze–thaw PMN could be as effective as freshly isolated human PMN. The numbers of animals used for control (CTRL), mtDAMPs, mtDAMPs + fresh PMN (Fresh PMN), and mtDAMPs + freeze–thaw PMN (F-T PMN) were n = 4, n = 7, and n = 8, and n = 8, respectively. Mean and SE values are shown. There was no significant difference by one-way ANOVA, Tukey.

## Data Availability

No new data were created. We are happy to share any information to researchers interested in our work.

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
