# Peer review of "Can Neutrophils Prevent Nosocomial Pneumonia after Serious Injury?"

_ijms, 2023, doi:10.3390/ijms24087627_

Round 1

Reviewer 1 Report

This is a well thought and insightful research article. My suggestion is for the authors to include the possibility of the application of PMN in preventing pneumonia among patients with immune deficiency disease and syndrome. could such research be applicable to these group of patients in preventing/treating pneumonia?

Author Response

This is a well thought and insightful research article. My suggestion is for the authors to include the possibility of the application of PMN in preventing pneumonia among patients with immune deficiency disease and syndrome. could such research be applicable to these group of patients in preventing/treating pneumonia?

Thank you very much for your positive comments. Indeed, we presume that the PMN could be applied before patients develop pneumonia and hence be preventive. Most of our data is relevant to seriously injured trauma patients, however, we believe that this approach could be useful for cancer patients after chemotherapy, patients with immune deficiency, and elderly patients whose PMN are less functional compared to younger people. We will modify the manuscript according to the Reviewers suggestion. See Lines 130-132.

Reviewer 2 Report

The authors suggest the nosocomial pneumonia is a leading cause of critical illness and mortality among serious injured trauma patients. They propose that PMNs migrate toward the injury site by detecting mitochondria associated molecular patterns (mtDAMPs), and PMNs are not responsive to secondary infection including bacteria-infected lungs, leading to nosocomial pneumonia. They set up a mouse model to prove their perspective about the intratracheal application of exogenously isolated PMNs to therapy nosocomial pneumonia after a serious injury. Although this is an interesting approach, the cell-based therapy is not a good idea for pneumonia. I suggest that the authors should perform more experiments to demonstrate the PMNs desensitization effect in response to bacterial infection, and complete a article with detailed materials and methods including the complete information about the mouse model and PMNs isolation from mouse and human.

Author Response

The authors suggest the nosocomial pneumonia is a leading cause of critical illness and mortality among serious injured trauma patients. They propose that PMNs migrate toward the injury site by detecting mitochondria associated molecular patterns (mtDAMPs), and PMNs are not responsive to secondary infection including bacteria-infected lungs, leading to nosocomial pneumonia. They set up a mouse model to prove their perspective about the intratracheal application of exogenously isolated PMNs to therapy nosocomial pneumonia after a serious injury. Although this is an interesting approach, the cell-based therapy is not a good idea for pneumonia. I suggest that the authors should perform more experiments to demonstrate the PMNs desensitization effect in response to bacterial infection, and complete a article with detailed materials and methods including the complete information about the mouse model and PMNs isolation from mouse and human.

We thank the Reviewer for the comments. Nosocomial pneumonia is the leading cause of the critical illness and mortality among seriously injured trauma patients. Therefore, new strategies/approaches are needed. We have documented that PMN in vitro are desensitized by mtDAMPs and lose their ability to “sense” bacteria and confirmed this in a mouse model (Ref. 7, 14-16, 44). This is an ongoing project and we need more experiments to support a possible PMN clinical application. Importantly, we propose to apply PMN to trauma patients as a preventive measure, before they develop pneumonia or full onset of pneumonia. It seems that present advances do not reject a theory that a cell-based therapy could be beneficial. At the end of manuscript (Lines 330-333), we listed list of  publications that explain detailed methods and knowledge behind perspective. We also added some figures for mouse models, which may help readers  understand/visualize our models better (Figures 1 and 2).

All changes can be found in yellow in the manuscript.

Reviewer 3 Report

The authors found that up to 28 days after human PMN application, they could not detect any lung injury, mainly due to the short life of PMN. They suggest that the timing of PMN migration to the lungs is critical in killing bacteria. The authors propose that PMN migration to the lungs may not occur when the bacterial infection starts due to receptor desensitization, but only after overwhelming bacterial growth when mtDAMPs in circulation finally decrease. They believe that the intratracheal application of PMN, including human origin, can effectively clear bacteria in the lungs without causing any adverse effects to the recipient. The authors are now testing the same concept in pig models as a step toward human use. They also question the importance of HLA match in PMN application, suggesting that PMN without HLA may be a safe candidate for PMN application once they are proven to function similarly to freshly isolated PMN.

More background about PMN need to be added, like how many hours postinjury will the PMN migrate to the injury site?

Title 2 and 3 are both vision, should be more specific.

Line 84, it is not clear how safe it is to apply XXX

Line 85, should be “But what important is xxxx”;

Is there well-established method to reverse the trauma patients’ PMN to the functional cells?

Line 161, injury followed by PMN treatment will be less confused.

Figure One, there are three groups: CON(non-injury, infected mice), mtDAMPs (injury, infected mice), mtDAMPs+PMN(injury, infected followed by treatment mice). I think a fourth group will offer more information for the comparison: non-injury, infected and followed by treatment mice. A one-way ANOVA (analysis of variance) would be appropriate if you want to test for differences in the mean number of SA in the lungs between the control group (CON) and mtDAMPs. A two-way ANOVA would be appropriate if you want to test for differences between CON and mtDAMPs+PMN for both the main effects of injury and PMN treatment and their interaction on the mean number of SA in the lungs. This test would determine if there is a significant difference in the mean number of SA in the lungs between the groups due to injury and/or PMN treatment, and if there is an interaction between injury and PMN treatment.

Author Response

The authors found that up to 28 days after human PMN application, they could not detect any lung injury, mainly due to the short life of PMN. They suggest that the timing of PMN migration to the lungs is critical in killing bacteria. The authors
propose that PMN migration to the lungs may not occur when the bacterial infection starts due to receptor desensitization, but only after overwhelming bacterial growth when mtDAMPs in circulation finally decrease. They believe that the intratracheal application of PMN, including human origin, can effectively clear bacteria in the lungs without causing any adverse effects to the recipient. The authors are now testing the same concept in pig models as a step toward human
use. They also question the importance of HLA match in PMN application, suggesting that PMN without HLA may be a safe candidate for PMN application once they are proven to function similarly to freshly isolated PMN.

More background about PMN need to be added, like how many hours postinjury will the PMN migrate to the injury site?

We thank the Reviewer for the comments. We have included additional information about PMN in the manuscript/background. The PMN, depending on the severity
and type of injury, may migrate into such spots really fast, in seconds or minutes.

Title 2 and 3 are both vision, should be more specific.

We agree with the Reviewer suggestion, so we modified the titles and the structure of the manuscript.

Line 84, it is not clear how safe it is to apply XXX.
Line 85, should be “But what important is xxxx”

We thank the Reviewer for noticing this part, which is not clear. We have modified these sentences accordingly.

Changes can be found in yellow highlight.

Is there well-established method to reverse the trauma patients’ PMN to the functional cells?

There is no established method so far. That is why we are working on it and we already have achieved some progress.

Line 161, injury followed by PMN treatment will be less confused.

Thank you, we have modified this part according to the Reviewer suggestion.

Figure One, there are three groups: CON(non-injury, infected mice), mtDAMPs (injury, infected mice), mtDAMPs+PMN(injury, infected followed by treatment mice). I think a fourth group will offer more information for the comparison: non-injury, infected and followed by treatment mice. A one-way ANOVA (analysis of variance) would be appropriate if you want to test for differences in the mean number of SA in the lungs between the control group (CON) and mtDAMPs. A two-way ANOVA would be appropriate if you want to test for differences between CON and mtDAMPs+PMN for both the main effects of injury and PMN treatment and their interaction on the mean number of SA in the lungs. This test would determine if there is a significant difference in the mean number of SA in the lungs between the groups due to injury and/or PMN treatment, and if there is an interaction between injury and PMN treatment.

Thank you for this suggestion. Indeed, this fourth group would be a good experimental addition. However, the data presented in Figure 1 have been already published hence the supplementary experiments would be difficult to perform in a timely manner. By using only these 3 groups we were still able to conclude that: 1. Injury (represented by application of mtDAMPs i.p.) reduces bacterial clearance and 2. The application of exogenous PMN to the lungs after injury decreases bacteria level to that observed in a control group (uninjured + bacterial infection). Therefore, we believe that data obtain with these three groups gave us enough information to support our hypothesis of beneficial effects of exogenous PMN. Regarding statistical analysis, as we mentioned earlier, these data have been  already published, with the statistical analysis approved there. Therefore, with your agreement, we would prefer to keep it this way, though, as you suggest, not the best tests were applied.

Round 2

Reviewer 2 Report

I have no additional suggestions or comments.